# Recent Progress in the Identification of Early Transition Biomarkers from Relapsing-Remitting to Progressive Multiple Sclerosis

**DOI:** 10.3390/ijms24054375

**Published:** 2023-02-22

**Authors:** Smaranda Maier, Laura Barcutean, Sebastian Andone, Doina Manu, Emanuela Sarmasan, Zoltan Bajko, Rodica Balasa

**Affiliations:** 1Ist Neurology Clinic, Emergency Clinical County Hospital Targu Mures, 540136 Targu Mures, Romania; 2Department of Neurology, ‘George Emil Palade’ University of Medicine, Pharmacy, Science, and Technology of Targu Mures, 540136 Targu Mures, Romania; 3Doctoral School, ‘George Emil Palade’ University of Medicine, Pharmacy, Science, and Technology of Targu Mures, 540142 Targu Mures, Romania; 4Center for Advanced Medical and Pharmaceutical Research, ‘George Emil Palade’ University of Medicine, Pharmacy, Science, and Technology of Targu Mures, 540136 Targu Mures, Romania

**Keywords:** multiple sclerosis, neurodegeneration, relapsing-remitting, secondary progressive, progression, biomarker

## Abstract

Despite extensive research into the pathophysiology of multiple sclerosis (MS) and recent developments in potent disease-modifying therapies (DMTs), two-thirds of relapsing-remitting MS patients transition to progressive MS (PMS). The main pathogenic mechanism in PMS is represented not by inflammation but by neurodegeneration, which leads to irreversible neurological disability. For this reason, this transition represents a critical factor for the long-term prognosis. Currently, the diagnosis of PMS can only be established retrospectively based on the progressive worsening of the disability over a period of at least 6 months. In some cases, the diagnosis of PMS is delayed for up to 3 years. With the approval of highly effective DMTs, some with proven effects on neurodegeneration, there is an urgent need for reliable biomarkers to identify this transition phase early and to select patients at a high risk of conversion to PMS. The purpose of this review is to discuss the progress made in the last decade in an attempt to find such a biomarker in the molecular field (serum and cerebrospinal fluid) between the magnetic resonance imaging parameters and optical coherence tomography measures.

## 1. Introduction

Multiple sclerosis (MS) is a chronic, inflammatory neurodegenerative disorder of the central nervous system (CNS) that represents the leading cause of non-traumatic disability in young adults. Despite extensive research into the pathophysiology of the disease in recent years, the underlying mechanisms have yet to be unravelled. The process of neurodegeneration was considered to occur secondarily to inflammation and demyelination, a consequence of an evolutive progression. However, recent data have confirmed that neurodegeneration is present in all clinical phenotypes, including early disease. Therefore, it was hypothesised that neurodegeneration and axonal loss are driven by independent pathological processes occurring before, or concomitant to, demyelination [1,2,3,4,5,6].

In 85–90% of cases, MS presents a relapsing-remitting course and is defined as relapsing-remitting MS (RRMS). RRMS is characterised by episodic occurrence of symptoms with complete or partial recovery followed by periods of clinical stability and remission. According to the natural history data of MS, 25% of RRMS patients eventually convert to progressive MS (PMS) after one decade of evolution. This progression is characterised by a gradual decline in neurological function with or without superimposed relapses. Due to recent developments in potent disease-modifying therapies (DMTs) and high-accuracy diagnostic protocols, the percentage of patients who transition to a progressive phase has decreased. Nonetheless, two-thirds of RRMS patients will progress to PMS after 20 years of disease evolution. The conceptual delineation between RRMS and PMS is not clearly defined; the evolution is gradual, with an intermediate phase of ‘transitional MS’. Compensatory mechanisms may initially protect against neuronal injury; however, in time, their continuous decline progresses to exhaustion and irreversible neurological deficits, marking the transition to PMS. However, the presence of these compensatory mechanisms represents one of the main obstacles in early PMS diagnosis. MS patients have a substantial heterogeneity concerning the brain reserve, modifiable cognitive reserve and neuroplasticity [7,8,9,10,11,12,13,14,15,16,17,18].

In 2014, a new definition by the MS Phenotype Group classified MS as relapsing-remitting and progressive disease. The early identification of the MS phenotype in which the predominant pathophysiological process is either inflammation (in RRMS) or neurodegeneration (in PMS) has a substantial impact on the therapeutic decision and the prognosis [18,19]. 

Neurodegeneration, independent of the inflammatory response, although present even in the early, asymptomatic stages of MS, represents the primary pathogenic mechanism in PMS and is responsible for the irreversible progression of disability, independent of relapses. A PMS diagnosis is exclusively retrospective and based on disability accumulation over 6 months to 1 year in the case of a patient who initially had a relapsing-remitting evolution of the disease. Currently, it is considered that the best definition of PMS is an increase by at least one point in the Expanded Disability Status Scale (EDSS) for patients who initially had an EDSS score ≤5.5 points or 0.5 points if the initial score was ≥6.0 points. The other conditions that must be fulfilled in order to meet the diagnostic criteria for PMS are the presence of an EDSS score of at least four points and involvement of the pyramidal system (functional score ≥2). Identifying the transition between RRMS and PMS according to the moment between the onset of disease progression and PMS diagnosis is difficult, and studies report an approximate 3-year delay in PMS diagnosis in 70% of cases. A delay in PMS diagnosis is associated with disability progression, alteration of the neurological status and a decrease in quality of life. Additionally, the financial burden is represented by the high reimbursement required for highly efficient DMTs in RRMS that hold no therapeutic value in PMS. Until recently, there was no incentive to select patients who were already in the transition phase due to the lack of an effective treatment for this form of MS. With the approval of siponimod in March 2019 as the first efficient treatment for active PMS, it became imperative to identify reliable and objective biomarkers of transition that could facilitate early diagnosis of PMS in the preclinical stage. This would enable early therapy adaptation (including timely escalation) to prevent PMS conversion and the irreversible progression of disability. In addition to the immunomodulator effect on the peripheral immune cells, siponimod is able to cross the blood–brain barrier (BBB) and interact with sphingosine 1-phosphate receptors inside the central nervous system, exerting a neuroprotective effect in PMS [7,8,13,20,21,22,23,24,25]. 

In the last 2 years, considerable progress has been made in identifying a feasible biomarker for early transition to PMS.

According to Lublin et al. [18], the diagnosis of PMS can be established only retrospectively, and the authors underline the lack of clinical, imaging, immunological and pathological biomarkers for the transition point from RRMS to PMS, and the pressing need to identify biological biomarkers that can enhance the MS phenotype definition. The authors suggested using optical coherence tomography (OCT) as an indirect biomarker for whole-brain tissue loss and the importance of additional studies aiming to extend the indications for OCT as a potential biomarker of MS phenotype. They suggest prospective follow-up for MS patients and their clinical and paraclinical parameters that should be prioritised to understand the disease’s subtypes better and to accurately predict the transition moment between RRMS and PMS. 

Recently, a Delphi panel in Portugal recognised that the identification of progression is a clinical challenge that would be completed by the discovery of biomarkers and diagnostic tools that point to the transition phase. The authors suggested that neurofilament light chains and OCT parameters hold value as biomarkers for disease progression [23]. 

This review aims to discuss the progress made in identifying biomarkers (molecular and imaging) that have prognostic value in the evolution of disability and the conversion of RRMS to PMS (Figure 1).

## 2. Molecular Biomarkers

### 2.1. Neurofilaments

Identifying an early serum biomarker for transition in MS patients that has value in current clinical practice is challenging. Most experimental studies were proven invalid due to a lack of specificity, sensibility and ease of sample harvest. Neurofilaments are major components of the neuronal cytoskeleton and are released into the extracellular space, cerebrospinal fluid (CSF) and bloodstream as a result of neuro–axonal lesions that appear in various pathologies of the central and peripheral nervous systems [26,27]. Three subunits have been described: NfL, neurofilament medium and neurofilament heavy chain (NfH). Since their discovery in 1989, the assessment of NfLs has proven effective for quantifying central neurodegeneration. Rosengren et al., demonstrated that Alzheimer’s disease and amyotrophic lateral sclerosis patients have significantly higher CSF NfL levels than healthy subjects, suggesting that CSF NfL titre can be used as a biomarker of neurodegeneration [28]. 

Further studies demonstrated that CSF NfL levels increase during MS relapses and are positively correlated with disability index and radiological activity. This evidence supports the hypothesis that CSF NfL can be used as biomarkers for treatment response [29,30,31,32]. Even though NfL concentrations are higher in the CSF than in the serum, the invasiveness of lumbar puncture greatly limits the applicability of a CSF examination in current practice.

However, this limitation was overcome with the introduction of the state-of-the-art single-molecule array (SiMoA) technique, which allowed for more precise assessment of low serum NfL concentrations. The feasibility of serum assessment using harvested peripheral blood allowed for the routine use of NfL assessment in clinical practice. The potential of NfL as a biomarker of disease progression emerged as a result of broad studies on large cohorts of MS patients. One of the most important studies was conducted by Disanto et al. The authors included 142 MS patients diagnosed with clinically isolated syndrome (CIS), RRMS, secondary-progressive (SPMS) and primary progressive MS (PPMS). In the cohort, serum NfL levels correlated with the duration of the disease and were significantly higher in SPMS and PPMS patients compared with CIS and RRMS patients. Univariate analysis revealed a significant association between NfL serum levels and EDSS (*p* < 0.001), recent decline in EDSS (*p* = 0.003) and relapse within the 2 months before harvest (*p* < 0.001). Furthermore, the authors evaluated the potential predictive role of NfL in predictive future clinical disease activity. They noted that in the first 2 years after NfL assessment, the incidence of clinical relapses was twofold higher in patients with NfL serum levels above the 97.5th percentile (IFF = 1.94, 95% CI = 1.21–3.10, *p* = 0.006 for the first year and IRR = 1.96, 95% CI = 1.22–3.15, *p* = 0.005 for the second year). Serum NfL levels correlated with disease worsening (calculated by the EDSS) at 12 months; 6.7% of the patients who initially had NFL levels below the 80th percentile presented with disease worsening, compared with 15% of the patients who had baseline NfL levels above the 97.5th percentile. These complex results support the role of NfL as an early biomarker for disease progression [26].

Siller et al., evaluated the role of NfL in neuronal destruction. They included 74 recently diagnosed CIS and RRMS patients, with a mean time from disease onset to serum sampling of 1 month; 63 of the patients were naïve to DMTs. The patients were assessed for NfL serum level at baseline and underwent 3T cerebral magnetic resonance imaging (MRI). Furthermore, 42 of the patients underwent 1–3 additional cerebral MRI assessments over the next 6–37 months. They demonstrated that baseline NfL serum levels significantly correlated with T2 lesion volume (T2-LV) and T2-LV progression throughout the study. Patients who presented with higher baseline NfL level had a higher T2-LV at 2-year follow-up, with an increase in cerebral volume loss in the 6–37 months following the initial MRI assessment. The accuracy of the cerebral atrophy prediction was higher for the 2-year follow-up cerebral MRI than the 1-year follow-up [33]. The results of this study further support the use of NfL as an early biomarker for progression in MS, before apparent imaging structural lesions appear.

The role of NfL in the pathophysiology of MS has been extensively studied. Clinical studies have demonstrated the utility of NFLs as a predictive biomarker for the progression of disability, brain and spinal cord atrophy, and cognitive decline [34,35,36,37]. A study by Barro et al., included 259 MS patients (189 with RRMS, 70 with PMS) and 259 healthy controls (HCs). The patients and HCs were assessed based on brain and spinal MRI, clinical data, and serum NfL over a mean period of 6.5 years. The results favoured NfL level as an independent predictive factor for disability progression in the next year (*p* < 0.001), and increased EDSS was directly correlated with NfL serum level. In addition, higher NfL levels were associated with more severe subsequent cerebral and spinal cord atrophy; at NfL values above the 97.5th percentile, an additional loss of 1.5% of the brain parenchyma and 2.5% of the spinal volume was observed over a period of 5 years [38].

Other longitudinal studies that followed patients with MS for one decade from the baseline assessment of serum NfL level, such as those performed by Thebault et al., and Chitnis et al., demonstrated the predictive role of NfL for the progression of disability, the risk of evolving towards a progressive form of MS, cerebral atrophy and increasing volume of T2 lesions [39,40].

Based on the reported evidence, NfL level was used in clinical studies to monitor treatment response. The results of a post-hoc analysis of the data compiled from two phase 3, placebo-controlled clinical trials published in 2022 by Leppert et al., supported the use of NfL as a biomarker of disease progression. A total of 1452 SPMS and 378 PPMS patients were included. A high baseline NfL level was associated with a significant increase in the risk of confirmed disability progression at 3 and 6 months, early wheelchair restriction and cognitive decline [41]. 

Serum NfL assessment using high-quality laboratory techniques, such as the SiMoA, represents a simple quantification method for neuronal damage. NfL assessment can be routinely used in clinical practice as a prognostic biomarker for disease progression.

Numerous studies demonstrated the utility of NfL as a prognostic biomarker in MS but, in contrast, NfH and phosphorylated NfH have been understudied. NfH and pNfH are cytoskeleton proteins expressed by the neurons in reaction to injury. They are of interest in neurodegenerative disorders such as amyotrophic lateral sclerosis, Alzheimer’s disease and frontotemporal dementia. Still, they have been studied to a lesser extent in MS. A recent study evaluated the role of serum/CSF NfH and OCT parameters in MS patients. The authors demonstrated that SPMS patients had significantly higher NfH CSF levels and lower thickness of the peripapillary retinal nerve fibre layer (pRNFL) compared with RRMS patients and controls (*p* < 0.0001). The pRNFL thickness positively correlated with serum and CSF NfH for both groups (*p* < 0.01). The EDSS scores positively correlated with the CSF NfH levels both in RRMS and SPMS groups (*p* < 0.001, *p* = 0.002) [42]. 

The importance of NfH levels in MS patients is disputed. While some authors demonstrated that the serum and CSF NfH levels are higher in PMS patients compared with RRMS patients, others disproved any association of this biomarker with a disability, progression risk and imaging parameters [43,44,45,46,47,48,49,50,51,52,53]. 

Due to the lack of consistency in the research results, the utility of NfH as a biomarker of transition in MS has yet to be established and further validation is needed. 

### 2.2. Circulating microRNAs

MicroRNAs (miRNAs) are short, non-coding, endogenous RNA molecules consisting of 21–25 nucleotides that are mainly involved in regulating gene expression, especially at the posttranscriptional level, by binding to target miRNAs, which leads to either degradation or translational repression [54,55].

MiRNAs have an essential role in tissue development and homeostasis, immune system up-regulation and maturation, and were shown to be involved in the pathophysiology of autoimmune disorders, such as MS [56,57,58,59]. Over 1500 human miRNAs regulating one-third of human genes have been identified [56]. Because they are susceptible to underlying pathological processes, as important molecules involved in cellular function regulation, and because of their stability, resistance to endogenous RNases, and easy assessment, circulating miRNAs have been proposed as biomarkers for early transition to PMS in RRMS patients [56]. 

Clinical studies have demonstrated the variable expression of certain miRNAs between RRMS and PMS (Table 1), suggesting that these miRNAs can be used as early biomarkers of conversion from RRMS to PMS. Haghiki et al., in a study that enrolled 53 MS patients (RRMS, SPMS, and PPMS) and 39 patients with other neurological pathologies, identified 50 miRNAs in the CSF. They compared the expression of five selected miRNAs (miR-132, miR-181c, miR-494, miR-633, miR-922) between the two groups of patients and between MS phenotypes. They reported that miR-633 and miR-181c exhibited significantly lower expression in the CSF of patients with SPMS compared with patients with RRMS. Following statistical analysis of the combination of the two miRNAs, a high sensitivity and specificity (69% and 82%, respectively) for differentiating between the two major types of MS was found [60]. The same researchers aimed to validate their results in a larger cohort of patients with MS (n = 218) and various other neurological pathologies (n = 211). They demonstrated that the same two miRNAs (miR-633 and miR-181c) were expressed differently in MS patients; however, the differences were not sufficient to differentiate RRMS from PMS patients [61]. 

Ghandi et al., aimed to identify the specific miRNAs present in different MS evolutive phenotypes. They included RRMS, SPMS and PPMS patients in the study and evaluated the expression of 368 miRNAs. The expression of eight miRNAs was significantly different between RRMS and SPMS patients: hsa-miR-92a-1, hsa-miR-135a, hsa-miR-454, hsa-miR-500, hsa-miR-574-3p, hsa-let-7c, hsa-let-7d and hsa-miR-145. Of these miRNAs, miR-92a-1 had the strongest association with RRMS (adjusted OR: 1.35, *p* = 0.022). MiR-92a-1 expression also correlated with the EDSS and the duration of the disease; furthermore, miR-92a-1 is a part of the miR-17-92 miRNA cluster that targets genes involved in cell cycle regulation, signalling, modulation and activation of naïve CD4+ T lymphocytes [62,63,64]. 

Recent imaging studies have demonstrated that cortical lesions are specific to PMS; however, conventional MRI sequences have technical challenges associated with their assessment. For this reason, Tripathy et al., aimed to identify miRNAs that could be used as predictive biomarkers of cortical involvement in MS. They evaluated 27 miRNAs that were differentially expressed in grey matter (GM) lesions (GMLs) compared with normal-appearing grey matter (NAGM), of which 10 were up-regulated and 17 were down-regulated [65]. By comparing their results with those of Fritsche et al. [66], who identified 31 miRNAs that were differentially expressed in cortical demyelinating lesions compared with HCs, the authors identified 4 miRNAs with specific expression in GMLs in both studies: miR-1180-3p, miR-219-a-2-3p, miR-328-3p and miR-432-5p. The results of this study were compared with the analysis performed by Regev et al. [67], who identified 87 serum miRNAs that correlated with the degree of cortical GM atrophy. Four of these miRNAs also had modified expression in GMLs in a study by Tripathy et al.: has-miR-149, has-miR-20a, has-miR-25 and has-miR-29c [65]. Furthermore, in Keller et al.,’s study [68], these four miRNAs were found in the peripheral circulation of MS patients and were specifically expressed by immune cells. These results support the use of these four miRNAs detected in MS patients’ serum as biomarkers of PMS, because of their specific expression in cortical demyelinating lesions and their association with cerebral atrophy. 

In a study of 120 MS patients, Regev et al. [67] aimed to identify the correlations between different selected miRNAs and MRI parameters (i.e., T2-LV, T1/T2 ratio lesion volume, global brain atrophy, GM atrophy, cervical spinal lesions and cervical spinal cord atrophy). Statistical analysis identified two miRNAs (has-miR-375 and has-miR-629-5p) that significantly correlated in both groups with cerebral atrophy, and two miRNAs (miR-486-5p and miR-92a-5p) that significantly correlated with the ratio of T1/T2 lesion volumes. The miRNAs associated with T2-LV (inflammatory and demyelinating processes) differed from those associated with cerebral atrophy (neurodegenerative processes). This observation supports the hypothesis that different pathological processes drive inflammation and neurodegeneration in MS.

In another study, Regev et al., proposed identifying circulating miRNAs that could be used to differentiate between RRMS and PMS phenotypes. The authors analysed 652 miRNAs to determine which were associated with inflammation and which were associated with neurodegeneration. In the discovery phase, by comparing the expression of miRNAs between RRMS and SPMS patients, 27 miRNAs with distinct expression between the 2 groups were identified. However, in the validation phase, of the 27 proposed miRNAs, only the expression of miR-27a-3p was significantly different (area under the curve value of 0.78) [69]. Additional studies demonstrated that miR-27a-3p inhibits the differentiation of Th1 and Th17 lymphocytes, is involved in the regulation of the neurotrophin pathway and is highly expressed in RRMS compared with SPMS patients [69,70]. These studies suggest that miR-27a-3p could function as a biomarker of transition from RRMS to PMS. In the described study, the expression of 10 miRNAs correlated with the degree of disability, and the highest correlation was for miR-199a-5p [69]. 

The same authors identified another miRNA as a potential biomarker of disease progression in MS patients: miR-337-3p. The expression of this miRNA was negatively correlated with disability (calculated by EDSS) in both the discovery and validation phases. Furthermore, they noticed that miR-337-3p expression was significantly lower in patients with SPMS compared with patients with RRMS (*p* = 0.01) [71]. Decreased miR-337-3p expression was associated with increased T1/T2 lesion load [67]. miR-337-3p targets RAP1A, a key component of chemokine-induced integrin activation and migration, which suggests that it can be used as a biomarker for natalizumab treatment response due to the drug’s mechanism of action, which involves inhibition of α4β1-integrin [67,71]. 

Sharaf-Eldin et al., aimed to identify biomarkers that facilitate the differential diagnosis of immune-mediated neuro-inflammatory pathologies. They included patients with neuromyelitis optica spectrum disorder, systemic lupus erythematosus with neuropsychiatric manifestations and MS. Even though the purpose of the study was to identify the differences between miRNAs in the selected disorders, the researchers determined the serum expression of three miRNAs (miR-145-5p, miR-223-3p and miR-326-5p) and noted that miR-326-5p expression was increased in RRMS compared with SPMS (*p* = 0.018). Furthermore, different combinations of the three miRNAs allowed for more accurate differentiation between the two clinical MS phenotypes: miR-145-5p + miR-326-5p (*p* = 0.014), miR-223-3p + miR-326-5p (*p* = 0.027) and miR-145-5p + miR-223-3p + miR-326-5p (*p* = 0.005). However, because of the small cohort, the data require validation in a larger population [72]. 

Exosomes are cell-derived extracellular vesicles that contain various molecules, including miRNAs, and can cross the BBB. Ebrahimkhani et al., hypothesised that the pathophysiological processes associated with different stages of MS evolution should reflect different expression of serum exosomal miRNAs between patients with RRMS and SPMS. They identified nine miRNAs (miR-15b-5p, miR-23a-3p, miR-223-3p, miR-374a-5p, miR-30b-5p, miR-433-3p, miR-485-3p, miR-342-3p, miR-432-5p) that could be used to differentiate patients with relapsing MS from patients with progressive forms of MS; furthermore, they demonstrated that the combination of miR-223-3p + miR-485-3p + miR-30b-5p differentiated between the subtypes with an accuracy of 95%. The results of this study support the use of exosomal miRNAs for early determination of the conversion from RRMS to PMS [73].

**Table 1 ijms-24-04375-t001:** Selected studies regarding miRNAs’ role as biomarkers for differentiating RRMS and SPMS patients.

Study	Reported Study Population	Evaluated miRNAs	Selected Reported Results
Regev et al., 2016 [69]	Discovery phase: 7 RRMS, 9 SPMS, 10 PPMS and 20 HCs Validation phase: 29 RRMS, 19 SPMS, 10 PPMS and 30 HCs	652 miRNAs were measured in the discovery phase and 40 in the validation phase (serum)	miR-27a-3p and miR-376b-3p were the only miRNAs showing significantly different expressions between RRMS and SPMS. After multiple comparations, only miR-27a-3p remained significant with an AUC of 0.78.
Gandhi et al., 2013 [62]	Discovery phase: 10 RRMS, 9 SPMS and 9 HCs Validation phase: 50 RRMS, 51 SPMS and 32 HCs	368 miRNAs were measured in the discovery phase and 19 miRNAs in the validation phase (plasma)	hsa-miR-92a-1, let-7 family, hsa-miR-454 and miR-145 had different expressions in RRMS vs. SPMS patients.
Haghikia et al., 2012 [60]	Discovery phase: 10 MS and 10 with OND Validation phase: 17 RRMS, 30 SPMS, 6 PPMS, 20 patients with OND	760 CSF miRNAs were measured in the discovery phase and the validation phase: miR-132, miR-181c, miR-494, miR-633 and miR-922 (CSF)	Only miR-181c and miR-633 differentiated RRMS patients from SPMS patients with a specificity of 82% and a sensitivity of 69%.
Kramer et al., 2019 [61]	81 RRMS, 106 SPMS, 12 PPMS and 218 patients with OND	miR-181c, miR-633 and miR-922 (CSF)	miR-181c levels were significantly different between RRMS and SPMS patients (*p* = 0.036)
Regev et al., 2018 [71]	Discovery phase: 7 RRMS, 9 SPMS and 20 HCs Validation phase: 29 RRMS, 19 SPMS and 30 HCs Reproducibility phase: 24 RRMS, 18 SPM and 30 HCs Transportability phase: 91 RRMS, 33 SPMS, and 58 HCs	Discovery phase: 652 miRNAs Validation phase: 192 miRNAs Reproducibility phase: 73 miRNAs Transportability phase: 73 miRNAs (serum)	hsa-miR-337-3p is under-expressed in SPMS compared with RRMS.
Sharaf-Eldin et al., 2017 [72]	18 RRMS, 19 SPMS, 20 patients with OND and 23 HCs	miR-145, miR-223 and miR-326 (serum)	miR-326 had a significantly lower expression in SPMS patients compared with RRMS patients (*p* = 0.017).
Khedr et al., 2022 [74]	50 MS patients and 50 HCs	miRNA-22 (serum)	miRNA-22 had a significantly higher expression in SPMS patients compared with RRMS patients (*p* = 0.042).
Ibrahim et al., 2020 [75]	39 RRMS, 35 SPMS and 10 HCs	miR-300 and miR-450b-5p (serum)	miR-300 and miR-450b-5p expression were significantly lower in SPMS patients, regardless of the EDSS score, compared with RRMS patients with EDSS ≤5.
Ebrahimkhani et al., 2017 [73]	Discovery phase: 14 RRMS, 7 SPMS, 4 PPMS and 11 HCs Validation phase: 14 RRMS, 9 SPMS, 2 PPMS and 11 HCs	168 miRNAs from exosomes (serum)	Nine miRNAs were identified with a different expression between RRMS and progressive MS patients: miR-15b-5p, miR-23a-3p, miR-223-3p, miR-374a-5p, miR-30b-5p, miR-433-3p, miR-485-3p, miR-342-3p, miR-432-5p). A combination of miR-223-3p, miR-485-3p and miR-30b-5p showed a 95% accuracy rate in distinguishing progressive forms of MS from RRMS.
Magner et al., 2016 [76]	59 RRMS, 15 SPMS and 25 HCs	64 miRNAs miR-103, miR-191, miR−423 (whole blood EDTA)	hsa-miR-874, hsa-miR-99a-5p, hsa-miR-141-3p, hsa-miR-125b-5p, hsa-miR-342-3p had different expressions in SPMS patients who responded poorly to IFNβ1a compared with RRMS patients who responded poorly (*p* < 0.01).

Abbreviations: AUC: area under the curve; CSF: cerebrospinal fluid; HCs: healthy controls; IFNβ: Interferon beta; RRMS: relapsing-remitting multiple sclerosis; SPMS: secondary progressive multiple sclerosis; PPMS: primary progressive multiple sclerosis; OND: other neurological disorders.

### 2.3. Glial Fibrillary Acid Protein

Glial fibrillary acid protein (GFAP), which is predominantly expressed by astrocytes, is another promising biomarker for the transition from RRMS to PMS, even if the study results are controversial. GFAP is an intermediate astrocytic cytoskeletal protein and a well-established marker of astrogliosis that plays a vital role in astrocytic-neuronal cross-talk. Abdelhak et al., performed a study on 42 RRMS, 13 SPMS and 25 PPMS patients and 20 patients with non-inflammatory neurological disorders. They assessed GFAP and NfL levels in the CSF and serum. The CSF and serum levels of GFAP were significantly increased in progressive MS patients compared with RRMS patients and patients with non-inflammatory neurological disorders; however, after performing age-adjusted corrections, the statistical significance was lost. In the PPMS patients, serum GFAP strongly correlated with EDSS after age-adjusted corrections. Furthermore, CSF and serum GFAP levels positively correlated with NfL levels in the PMS group [77].

A meta-analysis by Sun et al., identified four studies in which CSF GFAP levels were assessed in RRMS and PMS patients. Three of these studies demonstrated that CSF GFAP levels were increased in PMS compared with RRMS [77,78,79,80,81,82], indicating GFAP as a potential biomarker of disease progression in RRMS patients. Ayrignac et al., analysed serum GFAP levels in a cohort of 11 RRMS and 18 PPMS patients. Serum GFAP levels were significantly higher in PPMS patients than in RRMS patients, and the results remained statistically significant after multivariate analysis that included patient age and duration of the disease. Furthermore, serum GFAP levels were negatively correlated with white matter (WM) and GM volume [83].

### 2.4. Kynurenines

Tryptophan is an essential amino acid and 95% of it is metabolised by the kynurenine pathway (KP). Kynurenines are molecules resulting from tryptophan metabolism by the KP. Kynurenic acid (KYNA), the most studied molecule produced by astrocytes, has neuroprotective effects. In contrast, quinolinic acid (QUIN), produced by microglia, exhibits neurotoxic effects. The duality of the two kynurenines resides in their effect on NMDA, AMPA and kainite receptors (KYNA is an antagonist, while QUIN is an agonist). In addition, KYNA inhibits the influx of calcium ions into the intracellular space and prevents the formation of reactive oxygen species, while QUIN has the opposite effect [84,85,86].

Lim et al., analysed the metabolomic KP profile in MS patients, including RRMS, SPMS and PPMS patients. The QUIN/KYNA ratio was significantly higher in PPMS and SPMS patients compared with RRMS patients, and was highly correlated with the EDSS (r = 0.62, *p* < 0.0001). RRMS patients had higher KYNA levels compared with SPMS and PPMS patients, and high QUIN levels were reported in PPMS patients. Of the 37 potential biomarkers assessed by the authors, 6 could be used as predictive biomarkers of MS subtype, with an accuracy of 83%. In order of relevance, these biomarkers were KYNA, QUIN, tryptophan, picolinic acid, fibroblast growth factor-basic and tumour necrosis factor α [86]. In a study by Aeinehband et al., SPMS patients had lower KYNA and tryptophan levels compared with RRMS patients [87]. The results of these studies confirm the critical role of KP in neurodegenerative processes and the potential of KYNA and QUIN as biomarkers for the prediction of conversion to PMS. These studies demonstrate that kynurenine metabolites have different expression patterns depending on the MS clinical phenotype, and they may be used to indicate the transition from RRMS to PMS.

### 2.5. Chitinase-3-like-1 and Chemokine C-X-C Motif Ligand 13

The exact role of chitinase-3-like-1 (CHI3L1) in MS pathophysiology has yet to be completely understood. Inside the central compartment, activated microglia and reactive astrocytes are the main producers of CHI3L1. Chemokine C-X-C motif ligand 13 (CXCL13) is a B lymphocyte chemoattractant involved in forming ectopic lymphoid follicles adjacent to the meninges.

Recent clinical studies demonstrated a significant association between the serum and CSF levels of CHI3L1 with disability accumulation, cognitive decline and disease progression in MS patients [88,89,90,91,92].

In a recent systematic review and meta-analysis, Floro et al. [93] evaluated the potential to use serum and CSF CHI3L1 levels as a biomarker for distinguishing different MS phenotypes. They identified 20 studies published over a period of 10 years (2010–2020). Five studies compared the serum/CSF CHI3L1 levels in RRMS, SPMS and PPMS. They noted that PPMS patients are significantly higher compared with RRMS patients, but no differences were noted between RRMS and SPMS patients [93].

Lamancova et al. [94] evaluated serum levels of sNfL, CXCL13 and CHI3L1, among other molecular biomarkers, and noted that they were significantly higher in PPMS and SPMS patients compared with RRMS. Furthermore, the CXCL13 levels correlated with the EDSS score. In the SPMS group, the authors noted a positive correlation between CHI3L1 and sNfL.

Similar results are reported by Gil-Perontin et al. [95]. The authors analysed CSF levels for NfL and CHI3L1 in MS patients. The CSF CHI3L1 levels were lower in RRMS patients compared with SPMS and PPMS patients, but reached statistical significance only between RRMS and PPMS patients. In the same study, the CSF CHI3L1 positively correlated with the EDSS score and proved to be an independent factor of prediction for EDSS worsening by one point (HR = 2.99, 95% CI = 1.27–7.07). The patients were followed for 10 years and 14 patients with RRMS converted to SPMS. The statistical analysis identified the CSF CHI3L1 levels as a predictive biomarker for conversion. The results of these studies support the hypothesis that NfL, CHI3L1 and CXCL13 can be used to identify different MS phenotypes and have the potential to become predictive biomarkers of transition from RRMS to SPMS.

Two types of parallel inflammation are specific to MS. The first type is typical for the relapsing-remitting forms and is characterised by B and T lymphocyte passage through the vulnerable BBB, mainly with the formation of white matter demyelinating lesions. The second type of inflammation, although present from the early stages of the disease, defines compartmentalised inflammation in the presence of an impenetrable BBB. This results in the formation of tertiary ectopic lymphatic follicles adjacent to the meninge and the perivascular Virchow Robin spaces, and is associated with cerebral and cerebellar subpial demyelinating lesions [96].

The results of the abovementioned studies suggest that an increase in CHI3L1 production is associated with the second type of inflammation. This hypothesis was confirmed by the study of Cubas-Núñez et al. [97], who analysed the CSF CHI3L1 levels in different MS patients and evaluated the CHI3L1 expression and the degree of inflammation and neurodegeneration found in the autopsy tissue of 22 MS patients. The CSF CHI3L1 level was significantly lower in PMS patients compared with RRMS and, furthermore, correlated with the EDSS score. At the level of chronic active lesions, CHI3L1 is mainly expressed by astrocytes and is associated with neurodegeneration [97].

The results of these studies suggest that CHI3L1 and CXCL13 can be used to identify different MS phenotypes, but can also be used as predictive biomarkers for the moment of transition to a PMS. Still, this hypothesis needs to be confirmed in a larger cohort.

### 2.6. Other Potential Serum and CSF Biomarkers

Tau protein is a key component of the cytoskeleton of both neurons and oligodendrocytes and is essential for axonal transport. Abnormally phosphorylated P-tau leads to cell deterioration and is a characteristic feature of neurodegenerative diseases [98,99]. Anderson et al. [99,100] reported the presence of these abnormally phosphorylated tau proteins in PMS and patients with early aggressive MS. Other studies compared CSF tau protein levels between patients with RRMS and PMS; however, no significant differences were observed between these two groups [101].

The studies that analysed S100B in patients with RRMS and SPMS did not report significant differences [101].

Contactin-1 is a member of the contactin family and is mainly expressed in the paranodal axonal domain. Contactin-1 is involved in myelin formation and its loss is associated with neuronal dysfunction and axonal loss. Contactin-2 is found in the juxtaparanodal domain and plays a key role in axonal growth and guidance [102]. Chatterjee et al. [103] reported that CSF contactin-1 and -2 levels were lower in RRMS and SPMS patients compared with controls; however, no significant difference was observed between RRMS and SPMS patients.

## 3. Optical Coherence Tomography Biomarkers

Impairment of the visual system in MS is frequent. One-third of patients present at the onset with optic neuritis (ON), and most MS patients develop visual dysfunction throughout the course of the disease [104,105]. Therefore, the visual system has been the focus of numerous studies aiming to unravel the pathophysiological mechanisms that govern MS because the techniques used to assess the optic pathways are easily accessible, non-invasive and can be used in routine clinical practice. These include imaging investigations, such as MRI and OCT, and electrophysiological investigations, such as visual evoked potentials (VEPs).

The current literature supports the hypothesis that the visual system is a mirror of the pathophysiological processes at the CNS level, both in terms of the acute focal lesions that would have as a model at the level of the visual system the acute episodes of NO, but also in terms of the neurodegeneration process that would correspond to chronic retinopathy, optic neuropathy and transsynaptic degeneration [34,106,107]. In recent years, clinical studies have demonstrated that OCT can be used as a predictive biomarker for disease progression as it facilitates analysis of the neurodegenerative processes in MS [108,109].

OCT is a modern imaging technique that has significantly improved over the past two decades. It is a non-invasive, fast, easy-to-use, high-resolution technique that uses the same technology as ultrasound. It allows for the assessment of the optic nerve axon via a retinal nerve fibre layer (RNFL) measurement and neuronal assessment via a ganglion cell layer (GCL) measurement, and can detect subclinical neurodegeneration [108].

In 2009, Blumenthal et al. [110] compared RNFL measurements obtained by OCT and histopathological examination and concluded that they were similar. The potential use of OCT parameters as biomarkers of transition to PMS was demonstrated in numerous clinical studies. RNFL and GCL thickness can be used to differentiate RRMS patients from those with progressive forms of MS who present with apparent axonal and neuronal loss [111,112,113,114,115]. In a study on 414 MS patients, Oberwahrenbrock et al. [115] demonstrated that SPMS patients without a history of ON had a significantly thinner RNFL (*p* = 0.007) than patients with RRMS. RNFL thickness can be used as a biomarker of an early transition from RRMS to SPMS. A selection of the most relevant studies that demonstrated the usefulness of OCT parameters in differentiating RRMS from PMS patients is presented in Table 2.

RNFL and GCL thickness is correlated with consecrated biomarkers of neurodegeneration, such as global irreversible disability (quantified by EDSS; Table 3), global and specific brain atrophy [116,117], cortical lesion volumes [118] and fluid biomarkers, such as NfL. Given that the GCL thickness is difficult to measure accurately, in most studies the thickness of the composite GCL and inner plexiform layer (GCIPL) was evaluated [111,118].

Longitudinal studies demonstrated that RNFL and GCL thickness were independent predictive factors for disability progression [34,119,120,121]. Martinez-Lapiscina et al. [122] performed a multicentric research that included 74 CIS, 664 RRMS and 141 PMS patients. EDSS was calculated for all patients at baseline followed by OCT assessment. The patients were observed between 6 months and 5 years, with a median of 2 years. Patients that had at baseline RNFL thickness <88 µm or <87 µm (depending on the OCT), with no history of ON, had a higher risk of disability progression from year 1 to year 3 of follow-up (90% increase in adjusted risk), and the risk increased 4-fold after year 3 of follow-up.

**Table 2 ijms-24-04375-t002:** Studies regarding selected OCT parameters’ role as biomarkers for differentiating RRMS and PMS patients.

Study	Reported Study Population	Selected Evaluated Parameters	Selected Reported Results
Yurtogullari et al., 2022 [123]	66 RRMS, 31 SPMS	RFNL GCIPL	RNFL and GCIPL were thinner in the SPMS group compared with the RRMS group in patients who had no previous history of ON.
Pisa et al., 2020 [124]	26 NMOSD 29 RRMS, 20 SPMS and 3 PPMS	RNFL GCIPL	RRMS patients meeting NEDA-3 criteria had no significant RNFL thinning during the follow-up (13 patients; –0.271 µm/year; 95% CI = –0.892 to 0.349 µm/year; *p* = 0.392). The opposite is valid for PMS patients with NEDA-3 criteria, which had a significant RNFL loss during the study (11 patients; –0.556 µm/year; 95% CI = –0.985 to −0.127 µm/year; *p* = 0.011). Regarding the evaluation of GCIPL, in stable RRMS patients no significant thinning of this layer was observed, but in the case of progressive MS patients, with both active and stable forms of the disease, a significant thinning of GCIPL was observed.
Pulicken et al., 2007 [114]	135 RRMS, 12 PPMS and 16 SPMS	RNFL MV	The average RNFL thickness was reduced in both ON and non-ON MS patients compared with controls. The PPMS and SPMS groups had a lower baseline RNFL thickness than the RRMS group. A statistically significant reduced mean MV was noted in the SPMS patients compared with RRMS and SPMS patients compared with controls.
Gelfand et al., 2012 [113]	45 CIS, 403 RRMS, 60 SPMS and 33 PPMS	RNFL MV	In patients that had more advanced stages of the disease, the RNFL thinning was progressively increased based on the MS type (CIS = 98.2 ± 8.4 µm, RRMS = 92.9 ± 13 µm, SPMS = 85.5 ± 14.3 µm, PPMS = 80.5 ± 15.4 µm).
Oberwahrenbrock et al., 2012 [115]	308 RRMS, 65 SPMS and 41 PPMS	RNFL MV	The thickness of RNFL was reduced in SPMS eyes compared with RRMS eyes (*p* = 0.007), while total MV was lower in SPMS and PPMS eyes compared with RRMS eyes (all *p* < 0.05).
Balk et al., 2014. [125]	140 RRMS, 61 SPMS, 29 PPMS and 59 BMS	RNFL GCC	RNFL and GCC thickness were significantly lower in SPMS patients with no prior history of ON than in RRMS patients.
Jankowska-Lech et al., 2019 [126]	26 RRMS 3 PPMS, 7 SPMS and 12 PRMS	RNFL	The RNFL thickness in the temporal quadrant in RRMS patients (98.82 µm ± 12.35) was higher compared with PMS patients (86.47 µm ± 11.76), *p* = 0.025.
Estiasari et al., 2021 [127]	25 RRMS, 7 SPMS, 22 HC	RNFL GCIPL	SPMS patients have a reduced GCIPL and RNFL thickness, except for the nasal quadrant, compared with RRMS patients.
Jakimovski et al., 2021 [128]	109 RRMS, 35 PMS	RNFL MV	There was no difference in macular volume found. During follow-up, progressive MS patients with disability progression had decreased RNFL thickness compared with stable, progressive MS patients (69.53 μm vs. 79.9 μm, *p* = 0.007). A reduction in average RNFL was associated with progressive evolution, increased age and ON history.
Uzunköprü C et al., 2021 [42]	30 RRMS, 16 SPMS and 29 HCs	RNFL	SPMS patients had lower RNFL thickness in each eye as compared with the RRMS patients (*p* < 0.001).
Cellerino M et al., 2021 [129]	101 RRMS and 79 PMS	RNFL GCIPL	Reduced RNFL and GCIPL thickness were found in PMS patients rather than RRMS patients (all *p* < 0.05). Using multivariate models in the RRMS group, a correlation between GCIPL and follow-up disability was proved (0.04 increase in the EDSS for each 1 μm decrease in the baseline GCIPL, *p* = 0.02).
Sotirchos et al., 2020 [130]	178 RRMS, 126 SPMS, 60 PPMS and 66 HCs	RNFL GCIPL	All MS patients, regardless of their subtype and ON history, compared with HCs, had reduced GCIPL and RNFL thickness. When comparing MS subtypes, the SPMS and PPMS patients had lower GCIPL and RNFL thickness compared with RRMS patients. In the subgroup with no previous history of ON, the GCIPL and RNFL were thinner in SPMS and PPMS patients relative to RRMS patients.
Eslami et al., 2020 [131]	14 CIS, 92 RRMS and 14 SPMS	RNFL VM	RNFL thickness did not significantly differ between MS subtypes. The SPMS patients had the lowest total MV thickness (*p* = 0.04).
Behbehani et al., 2017 [111]	84 RRMS, 27 SPMS, 2 PPMS and 38 HCs	RNFL GCIPL	Statistically significant differences were found in PMS compared with RRMS patients regarding RNFL (*p* = 0.02) and GCIPL (*p* = 0.006), with lower values in the first group.

Abbreviations: BMS: benign multiple sclerosis; CIS: clinically isolated syndrome; CI: confidence interval; EDSS: Expanded Disability Status Scale; GCC: ganglion cell complex; GCIPL: ganglion cell inner plexiform layer; HCs: healthy controls; MV: macular volume; NMOSD: neuromyelitis optica spectrum disorders; NEDA: no evidence of disease activity; ON: optic neuritis; OR: adjusted odds ratio; PPMS: primary progressive multiple sclerosis; PMS: progressive multiple sclerosis; PRMS: progressive relapsing multiple sclerosis; RNFL: retinal nerve fibre layer, RRMS: relapsing-remitting multiple sclerosis; SPMS: secondary progressive multiple sclerosis.

**Table 3 ijms-24-04375-t003:** Selected studies assessing correlations between OCT parameters and EDSS.

Study	Reported Study Population	Selected Reported Results
Berek et al., 2022 [132]	93 MS	Low baseline values of RNFL and GCIPL are associated with disability progression after 6 years. GCIPL thinning is associated with EDSS worsening.
Balıkçı et al., 2021 [133]	7 CIS, 51 RRMS, 21 SPMS, 4 PP and 57 HCs	EDSS negatively correlated with mean RNFL (*p* < 0.001) and GCC measurements (*p* < 0.001).
Piedrabuena et al., 2022 [134]	52 RRMS	A negative correlation between EDSS and RNFL was found. No other significant differences were found in the assessments for the 2-year follow-up.
Barreiro-González et al., 2022 [135]	64 MS	Patients’ EDSS values correlated with age (r = 0.543, *p* = 0.001), spinal cord volume (r = −0.301, *p* = 0.034) and GCL (GCL, r = −0.354, *p* = 0.012).
Bsteh et al., 2019 [136]	151 RRMS	In patients with RNFL thickness ≤88 µm, there was three times increased risk of EDSS progression (*p* < 0.001) and 2.7 times increased risk of cognitive decline in the next 3 years (*p* < 0.001).
Schurz et al., 2021 [137]	53 RRMS and 7 SPMS	When comparing to the stable group, the patients that had disability worsening had a reduced thickness of RNFL (83.4 µm vs. 97.7 µm, *p* = 0.001) and GCIPL (69.3 µm vs. 78.2 µm, *p* < 0.001), *p* = 0.672). Patients with GCIPL thickness <77 µm at baseline were associated with four times increased risk of disability worsening during the follow-up. Patients with RNFL ≤88μm had a weaker significant association with increased risk of disability worsening (HR 3.1; 95% CI = 1.4–7.0; *p* = 0.019.
Lambe et al., 2021 [121]	106 RRMS and 26 SPMS	In patients with an average baseline GCIPL <70 µm, there was an increased four times risk of EDSS worsening association (OR: 3.97, 95% CI = 1.24–12.70; *p* = 0.02). An independent association was found between a lower baseline GCIPL thickness and long-term disability worsening.
Cilingir et al., 2021 [138]	137 RRMS	Both intermediate (94–100 µm) and reduced (<94 µm) RNFL thickness are associated with an increase in EDSS progression in the first 2 years.
Bitirgen et al., 2020 [139]	31 RRMS	A significant reduction in RNFL thickness (*p* = 0.004) was associated with an increase in EDSS over 2 years.
Cilingir et al., 2020 [140]	134 RRMS	A significant correlation was found between EDSS progression and RNFL in both early-stage and late-stage RRMS patient groups (r = −0.471, *p* < 0.001, and r = −0.567, *p* < 0.001).
Bsteh et al., 2021 [141]	183 RRMS	An increased risk of disability progression was found in patients with baseline GCIPL thickness <77 μm (HR: 2.7, 95% CI = 1.5–4.7, *p* < 0.001). A strong predictor for clinical progression was the annual loss of macular GCIPL, with a cut-off value ≥1 µm, which accurately identified clinically progressive patients (87% sensitivity at 90% specificity, OR: 18.3, 95% CI = 8.8–50.3).
Vidal-Jordana et al., 2020 [142]	109 RRMS	Patients with a reduced RNFL thickness had associated decreased BPF, SC area values, a higher T2 lesion volume and a higher lesion volume within the optic radiations (all *p* < 0.05). Reduced GCIPL volumes were correlated with decreased BPF values and SC area values. Patients with RNFL thickness ≤86 µm had 4.7–6.7 times increased chance of reaching an EDSS ≥3.0. Also, patients with GCIPL volume ≤1.77 mm^3^ had a 9.7–11.9 times increased chance of reaching EDSS ≥ 3.0.
Koraysha et al., 2019 [143]	49 RRMS	RRMS patients with an EDSS >2, compared with those with an EDSS ≤ 2.0, had lower RNFL thickness (*p* = 0.01).
Cordano et al., 2018 [144]	228 RRMS, 29 SPMS, 32 CIS and 10 PPMS	A multivariate linear regression analysis was used to analyse the association between baseline RNFL and EDSS, adjusted by age and sex. It was found that every 1 µm decrease in the thickness of RNFL was associated with a 0.024 increase in EDSS (95% CI = 0.011–0.037; *p* < 0.001). Similar results were found in sensitivity analysis during a >5-year follow-up (0.02 increase in the EDSS, 95% CI = 0.03–0.01, *p* = 0.001).
Garcia-Martin et al., 2017 [145]	100 RRMS and 50 HC	Correlation between high EDSS scores and temporal and superior RNFL thickness reduction.
Albrecht et al., 2012 [146]	42 RRMS, 41 SPMS and 12 PPMS	Statistically significant correlations were found between mean RNFL and EDSS (r = −0.36, r = −0.35 and r = 0.33; all *p* < 0.05). The correlations remained significant even after excluding any previous ON history (r = −0.36, r = −0.35 and r = 0.33; all *p* < 0.05). EDSS had negative correlations with RNFL.
Bsteh et al., 2020 [147]	171 RRMS	There was no difference in RNFL and GCIPL thickness related to relapses or relapse-free during the follow-up. Multivariate regression analysis found an association between PIRA and the reduction of GCIPL and RNFL thickness.
Bsteh et al., 2019 [148]	141 RRMS	When comparing RRMS patients that progressed during the study with those that remained clinically stable, they found an increase in RNFL thinning in the progressing group (0.5 μm, *p* < 0.001). A cut-off value of RNFL >1.5 um was used to distinguish between stable and progressing RRMS (specificity 90%, sensitivity 76.1%). The exact cut-off value was associated with a 15-fold increase in the risk of clinically progressing MS (*p* < 0.001).
Martinez-Lapiscina et al., 2016 [122]	74 CIS, 664 RRMS, 83 SPMS and 58 PPMS	Patients with RNFL less than or equal to 87/88 μm had a double risk of disability worsening at any time from the first to the third year of follow-up (HR: 2.06, 95% CI = 1.36–3.11; *p* = 0.001). The risk increased almost 4 times from the 3rd to the 5th year of follow-up.
Rothman et al., 2019 [120]	151 RRMS, 14 SPMS and 7 PPMS	There is an association between lower baseline total MV and a higher 10-year EDSS score, which was shown in the multivariable models (mean increase in EDSS of 0.75 per 1 mm^3^ loss in total MV (*p* = 0.02). Patients in the lowest tertile of baseline total MV had an average higher EDSS score at 10 years (mean difference = 0.86; 95% CI = 0.23–1.48) and had a 3.5 times increased chance of clinically significant EDSS worsening when compared with the patients in the highest tertile of baseline total MV (OR: 3.58; 95% CI = 1.30–9.82; *p* = 0.008). Using univariate models, it was possible to use RNFL to predict the 10-year EDSS progression.
Lambe et al., 2021 [121]	92 RRMS and 25 PMS	The follow-up of the patients was an average of 10.4 years. Using multivariate models that excluded patients with prior ON, an average baseline GCIPL thickness <70 µm correlated with four times increased chance of significant EDSS worsening (*p* = 0.02) when compared with patients with an average GCIPL thickness ≥70 µm.

Abbreviations: BPF: brain parenchyma fraction; CIS: clinically isolated syndrome; CI: confidence interval; EDSS: Expanded Disability Status Scale; GCC: ganglion cell complex; GCIPL: ganglion cell inner plexiform layer; GCL: ganglion cell layer; HCs: healthy controls; HR: hazard ratio; MV: macular volume; NMOSD: neuromyelitis optica spectrum disorders; ON: optic neuritis; OR: adjusted odds ratio; PPMS: primary progressive multiple sclerosis; PMS: progressive multiple sclerosis; RNFL: retinal nerve fibre layer, RRMS: relapsing-remitting multiple sclerosis; SPMS: secondary progressive multiple sclerosis; PIRA: progression independent of relapses; PPMS: primary progressive multiple sclerosis; SC: spinal cord.

The results of these studies suggest that RNFL and GCIPL are promising OCT biomarkers of axonal loss and neurodegeneration and can be used to monitor disease progression and hold value as predictive biomarkers of transition to PMS.

## 4. Magnetic Resonance Imaging Biomarkers

The recent advances in neuroimaging, especially in MRI techniques, have drastically challenged the understating of basic pathophysiological processes in MS and have facilitated early diagnosis of neurodegeneration. Conventional MRI sequences allow for the assessment of WM demyelination and the identification of active lesions. Today, the leading cause of irreversible disability accumulation and cognitive decline is neurodegeneration, not inflammation. Due to the availability of new DMTs, it is imperative to identify imaging biomarkers with prognostic value for disability progression and conversion to PMS [149,150,151,152].

### 4.1. Global and Regional Brain Atrophy

For many years, the accumulation and expansion of new T2 lesions and contrast-enhancing T1 lesions (both biomarkers of inflammation) were the only parameters used for radiological follow-up and treatment-response monitoring in MS patients. However, these parameters do not correlate with the clinical evolution of the disease. Irreversible progression of disability is associated with neurodegeneration rather than inflammation, and it is highly associated with the degree of cortical atrophy [153,154,155,156]. Cortical atrophy is the most studied biomarker of neurodegeneration and represents one of the final sequelae of neurodegenerative processes. Novel MRI techniques allow for global and regional brain atrophy assessment from the early stages of the disease. However, they are unfeasible in current clinical practice due to technical limitations and physiological variations in cerebral volume (e.g., hydration, menstrual cycle) that require extensive knowledge of, and experience with, these imaging techniques. Other factors that influence the accuracy of the results are genetics (expression of apolipoprotein E), lifestyle (e.g., alcohol, smoking), concomitant disorders (diabetes and cardiovascular diseases) and medications [149].

Numerous clinical studies have demonstrated that the quantification of global brain atrophy can be a predictive biomarker for CIS conversion to RRMS [151,157]. Clinical progression of disability independent of relapses in RRMS patients, referred to as ‘silent progression’, is associated with a higher cerebral atrophy rate than in clinically stable RMS patients [158]. The role of cerebral atrophy as a biomarker for MS phenotype stratification, its association with disability severity and the presence and severity of cognitive decline were demonstrated in clinical studies. Therefore, cerebral atrophy assessment was added to the ‘no evidence of disease activity (NEDA)’ criteria 4 (NEDA-4) for the evaluation of DMT efficiency alongside other radiological (new T2 lesions/increasing T2 lesions, gadolinium-enhancing lesions) and clinical parameters (clinical relapses, confirmed disability progression) [149,159,160,161,162,163].

Global cerebral atrophy is present in all stages of MS. A study that followed 206 patients (180 RRMS, 14 SPMS, 12 PPMS) and 35 HCs demonstrated that the annualised percent brain volume change (PBVC/y) was significantly higher in MS patients (−0.51 ± 0.27%) than in HCs (−0.27 ± 0.15%). The annual brain volume loss rate exceeded the physiological limits in all disease phenotypes, with a PBVC/y of −0.52 ± 0.29% in RRMS patients and −0.45 ± 0.18% in PMS patients. ROC analysis established a PBVC/y cut-off of −0.37%, with a sensitivity of 67% and a specificity of 80% in distinguishing MS patients from HCs. When the cut-off was modified to −0.52%, the specificity reached 95% (i.e., the rate of false positives decreased to 5%). Even a cut-off of −0.4% had clinical significance; patients with a brain volume loss rate higher than −0.4% had a higher rate of EDSS decline [164].

Evaluating the segmental volume of different cerebral regions has been of interest in recent years. The development of new imaging techniques has allowed for a more detailed assessment of cortical GM and deep grey matter (DGM). Global brain atrophy appears secondary to neurodegeneration and neuroaxonal loss at the level of the GM. Patients with SPMS have reduced cortical GM and DGM volume compared with RRMS patients. Loss of GM volume in the temporal pole and posterior insula is more accelerated in SPMS patients (−1.21%) than in RRMS patients (−0.77%). DGM volume has a predictive value for disability accumulation in the time interval until progression if confirmed with EDSS. Eshaghi et al. [161] reported the highest predictive value of EDSS for progression in RRMS patients in thalamic volume (baseline thalamic atrophy increased the disability risk by 37%), hippocampal volume and angular gyrus volume. DGM atrophy appears early in MS evolution. A significant decrease in DGM volume was noted in CIS patients compared with HCs; however, no differences were reported regarding global brain volume and WM volume between the two groups. DGM atrophy evolves more rapidly than WM atrophy [149,161,163].

In addition to regional brain atrophy, isolated thalamic atrophy has the highest potential to be a prognostic biomarker for disease progression and disease progression. In a recent study by Hänninen et al. [163], the authors included 24 newly diagnosed RRMS patients and 36 SPMS patients. They demonstrated that isolated thalamic atrophy appeared before global brain atrophy (1/60 patients had global brain atrophy in the absence of thalamic atrophy and 16/60 patients had thalamic atrophy in the absence of global brain atrophy). SPMS patients had a significantly lower thalamic volume at baseline and at the end of the study than RRMS patients. Isolated thalamic atrophy at the onset of the study, even when not associated with global brain atrophy, represented a risk factor for EDSS progression and not reaching NEDA-3 at 2-year follow-up. The evaluation of global brain atrophy and atrophy of specific regions can be used to identify patients at risk of disease progression and transition from RRMS to PMS.

### 4.2. Chronic Active Lesions and Cortical Lesions

Recent pathological studies have demonstrated that the progressive neurodegeneration found in more advanced stages of MS occurs secondary to chronic inflammation that is compartmentalised into the CNS. When the BBB is intact, inflammatory infiltrates are primarily found in the meningeal and perivascular spaces [165,166,167]. Some demyelinating lesions undergo early remyelination after the initial inflammatory stage, which prevents axonal degeneration, while others remain chronically demyelinated, without the presence of the inflammatory infiltrate, but with axonal loss and gliosis, or ‘chronic inactive lesions’ [168]. A third category of WM lesions is represented by chronic active lesions, or so-called ‘smouldering lesions’ or ‘slowly expanding lesions’ (SELs). In terms of pathophysiology, SELs present a central inactive demyelinated core surrounded by a slow, continuous inflammatory demyelinating process at the periphery, overlapped with incomplete remyelination. The result is irreversible myelin loss and axonal degeneration. Ongoing myelin degradation is followed by phagocytosis, which occurs at the periphery of the demyelinating lesions. Iron accumulates in activated microglia, which induces a paramagnetic rim at the lesion’s edge consisting of iron-loaded microglia on susceptibility-weighted images (SWI), and is the reason why this type is also called an ‘iron rim lesion’ (IRL). One-half of WM lesions have persistent inflammation and function as active chronic lesions in relapsing MS patients and at least in 60% of progressive MS patients [168,169]. Frequently, these lesions slowly expand and progress rapidly and with increased severity; their microstructural abnormalities are reflected by a T1 hypo signal, and they represent promising biomarkers for the transition of RRMS to PMS [165,170,171,172].

Cortical lesions (CLs) are another promising biomarker of disease progression. CLs are present in all evolutive phenotypes of MS, but their burden is highest in SPMS patients. CL number and volume correlate with disability progression, cognitive decline and the degree of cortical and global brain atrophy. A 7T MRI is required to detect CLs; conventionally, most clinical sites have 3T MRIs. Given the current radiological techniques, their applicability as progression biomarkers in current clinical practice is minimal [173,174]. Calabrese et al. [175] performed a longitudinal prospective study over a period of 5 years that included RMS and PMS patients. The authors demonstrated that a high CL burden at baseline was associated with clinical progression of the disease throughout the study. Therefore, the number of baseline CLs is an independent predictive factor for the progression of disability.

Treaba et al. [170] performed a study on 111 MS patients: 74 with RRMS and 37 with SPMS. They used a 7T MRI scanner for radiological assessment. The number and volume of CLs in SPMS patients was significantly higher than in RRMS patients (*p* < 0.001). The model for disease-stage prediction reached a mean ± standard deviation (SD) for the area under the curve of 0.82 ± 0.08. The six most important radiological parameters that facilitated RRMS and SPMS distinction were global WM volume, thalamic volume, number and volume of WM lesions with no rim, leukocortical lesion volume and intraventricular CSF volume. In this model, the number of leukocortical lesions and IRLs was ranked low, as the 12th and 14th predictors of MS subtype; however, they were the main predictors of neurological disability progression (measured using the EDSS) over a mean period of 3.2 years. Therefore, IRLs and leukocortical volume have value as predictive biomarkers for disability progression, with salient implications on therapeutic strategies.

Evaluating the hypothesis that the number and characteristics of SELs are predictive biomarkers for disability progression and conversion to PMS, Preziosa et al., published a study of 52 RRMS patients who were prospectively followed for 9.1 years and underwent imaging tests at baseline and after 6, 12 and 24 months. A significant association was noted between EDSS progression at 9.1 years and the median proportion of SELs (*p* = 0.045), the presence of at least four SEL lesions (*p* = 0.018), and the mean baseline magnetisation transfer ratio of the SELs (*p* = 0.026). Regarding the conversion risk to SPMS at 9.1 years from the baseline, the data revealed a significant association only with a decreased SEL magnetisation transfer ratio at onset (*p* = 0.037) and a reduction in T1 SEL signal on the 2-year MRI compared with the baseline MRI (*p* = 0.041). The results of this study suggest that the proportion of SEL-type lesions and their microstructural alterations are independent risk factors for disability progression (measured using the EDSS) and conversion to SPMS [165].

Various other clinical studies demonstrated the association between chronic active lesions and the clinical evolution of MS. Absinta et al., demonstrated that patients with ≥4 IRLs at baseline had a 1.6-fold increased risk of developing clinical disease progression compared with patients with no IRLs at baseline. When patients over 50 years old were excluded from the statistical analysis, the risk of developing clinical progression increased to 3.2-fold. Moreover, patients with ≥4 IRLs had a higher degree of global and regional brain atrophy (thalamus, putamen, caudate nucleus) than patients with no IRLs [169]. In most clinical studies, SEL and IRL assessments were performed using a 7T MRI apparatus; however, in recent years, published studies have demonstrated that these lesions can be easily visualised and assessed using a 3T MRI with SWI. This gradient echo imaging technique facilitates the identification of different MS lesions based on their paramagnetic properties. Blinderbacher et al., followed a group of patients with CIS (n = 32) and RRMS (n = 34) over 2.9 years. Using the SWI sequence, the researchers noted that IRLs were present in early MS stages. In 13 (19%) patients, these lesions were found at baseline, and the subjects had significantly lower Symbol Digit Modalities Test scores compared with the non-IRL subjects. A high IRL index correlated with higher brain volume loss throughout the study. Moreover, baseline IRLs correlated with a higher EDSS during the study, and statistical analysis revealed that baseline IRLs were independent risk factors for future disability progression [176].

After Hemond et al., demonstrated that paramagnetic rim lesions can also be visualized on susceptibility-weighted sequences using 1.5 T MRI, chronic active lesions have become an important imaging biomarker with the potential to be used in current clinical practice to evaluate the evolution of MS patients and the risk of disease progression [177].

### 4.3. Spinal Cord Damage

As research advances on the identification of radiological biomarkers for the conversion from RRMS to PMS, in addition to conventional cerebral parameters, it has become apparent that spinal cord imaging has value as a predictor of progression. The annual rate of spinal cord atrophy is approximately −1.78% when both relapsing and progressive forms of MS are analysed and −2.08% when only progressive cases are analysed. The mean annual rate of cerebral atrophy is only −0.4% [178,179]. Spinal cord atrophy is less studied as a radiological parameter than cerebral atrophy in MS, mainly due to anatomical (smaller dimensions, higher flexibility) and radiological (lower tissue contrast, possible artefacts due to breathing patterns) limitations. Recent studies have demonstrated that three-dimensional sagittal T1-weighted sequences, which are part of routine cerebral MRI assessment in MS patients, include the upper part of the spinal cord, which is less affected by breathing patterns and pulmonary artefacts. Assessment of the spinal cord area is usually made at the level of C1–C2, C1–C3 and C2–C3 [149,180,181,182].

In a study published by Rocca et al., that followed 326 patients with RRMS, 41 with SPMS and 179 HCs for about 5 years, one-third of the patients presented with an increase in the EDSS, and 14% of the RRMS patients converted to SPMS. The low surface area of the spine and high baseline spinal cord lesions were independent predictive factors for disability in the next 5 years. Moreover, statistical analysis revealed that baseline spinal lesion burden was an independent risk factor for conversion to SPMS [183]. Bischof et al., followed an MS cohort (360 with RRMS and 47 with SPMS) and 80 HCs for a period of 12 years. They reported that 54 patients with RRMS converted to SPMS, and silent progression of disability was noted in 159 cases. The patients who had RRMS at baseline and converted to SPMS presented a high rate of cervical spinal cord atrophy, assessed at C1 vertebral level (−2.19%/year) at least 4 years before conversion, while the patients who remained diagnosed with RRMS had a mean rate of cervical spinal cord atrophy of −0.88%/year. After progressing to an SPMS phenotype, the spinal cord atrophy rate slowed to 1.63%/year. Statistical analysis revealed that the annual spinal cord atrophy rate was a predictive factor for disability progression and conversion to SPMS, as even a 1% decrease was associated with a reduced time interval (59%) to reaching the progressive phase of the disease. The patients who experienced silent progression of disability had an annual spinal cord atrophy rate of −1.1%/year, while the patients with clinical stability had a significantly lower rate of spinal area loss (−0.72%/year) (*p* = 0.004) [184].

Spinal cord atrophy is a valuable biomarker for predicting the progression of disability and progression to PMS.

### 4.4. Atrophied Lesion Volume

Early clinical progression of disability in RRMS patients towards PMS can be quantified by the atrophied T2-LV, which represents either the lesional tissue volume being replaced by CSF secondary to atrophy or the direct destruction of the lesion. In a study that included 176 RRMS patients who were followed clinically and radiologically for 10 years, the atrophied T2-LV in the first 6 months from baseline was used to differentiate between the patients who remained clinically stable throughout the study (n = 76) and the patients who presented with confirmed disability progression (n = 100). This is clinically important, as PBVC assessment could only be used to differentiate between the two groups after 2 years of radiological follow-up [149,153,185].

These results were confirmed in a much larger MS cohort (1089 RRMS, 217 SPMS, 124 CIS patients). Genovese et al., demonstrated that SPMS patients had a higher annualised atrophied T2-LV rate compared with RRMS patients (*p* = 0.001). In addition, the patients who were clinically stable throughout the entire study period had a significantly lower annualised atrophied T2-LV than the patients with confirmed disability progression (*p* < 0.001), and logistic regression analysis revealed that an annual increase of 1 mL of atrophied T2-LV was associated with a five-fold increase in the risk of disability progression. The patients who converted to SPMS (n = 67) had a higher annualised atrophied T2-LV compared with clinically stable patients (*p* = 0.002), and an increase of 1 mL per year of annualised atrophied T2-LV was associated with a 4.7-fold increased risk of developing SPMS [186]. Therefore, atrophied T2-LV is a potential radiological biomarker for early prediction of disability progression and disease conversion from RRMS to PMS.

## 5. Conclusions

This study aimed to review potential biomarkers of transition that could facilitate early identification of RRMS patients who are at risk of progressing to PMS. This knowledge would allow for prompt DMT adjustment or escalation to constrain and delay the conversion towards PMS. The clinical implications of the latter objective would shorten the time interval until a diagnosis of PMS is established, and an appropriate and prompt therapeutic decision could be made before irreversible disability occurs. Early identification of transition biomarkers to PMS could represent a key step in paving the way towards personalised therapy.

The molecular, OCT and MRI biomarkers presented in this review have promising value as predictive biomarkers of early transition from RRMS to PMS. The most promising paraclinical biomarkers that can feasibly be translated to current clinical practice are serum NfL, thalamic atrophy, chronic active lesions, spinal cord atrophy, GCIPL and RNFL; however, they need to be further validated in longitudinal, long-term studies with larger cohorts of MS patients.

## Figures and Tables

**Figure 1 ijms-24-04375-f001:**
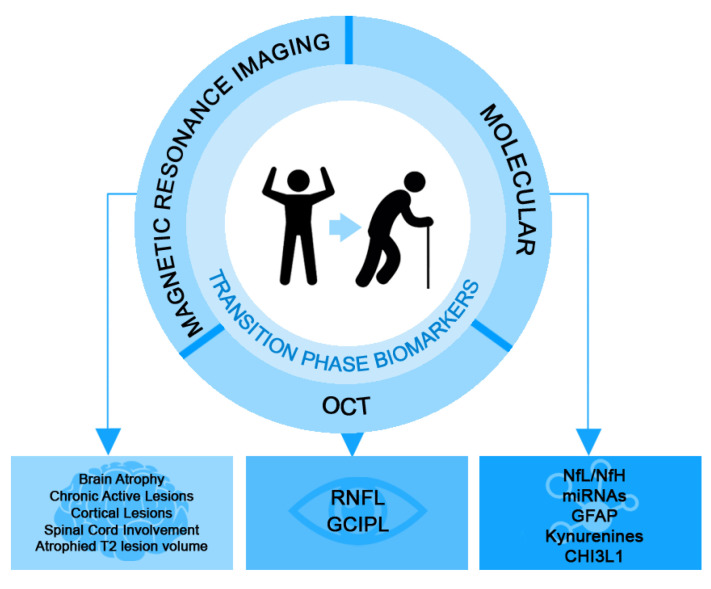
Proposed biomarkers with prognostic value for the conversion of RRMS to PMS (CHI3L1: chitinase 3-like-1; GFAP: glial fibrillary acidic protein; GCIPL: ganglion cell–inner plexiform layer; Nfl: neurofilament light chain; NfH: neurofilament heavy chain; OCT: ocular coherence tomography; RNFL: retinal nerve fibre layer).

## Data Availability

Not applicable.

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
