# Peer review of "Recent Progress in the Identification of Early Transition Biomarkers from Relapsing-Remitting to Progressive Multiple Sclerosis"

_ijms, 2023, doi:10.3390/ijms24054375_

Round 1
Reviewer 1 Report
I have a major concern and I believe the manuscript should be re-written according to this concern:
1- The classification of MS has been changed and the authors should use the newest classification which divided MS to relapsing and progressive forms.
2- According to studies neurodegeneration is ongoing process which begins at the onset of disease. So, term of transitional stage of this disease is old and it should be revised.
3- The biomarker related to transitional phase are not useful.
4- It is better to describe which biomarkers could predict the speed and amount of neurodegeneration in this patient.
Author Response
Esteemed reviewer,
On behalf of my co-authors and I, we would like to thank you for considering our article for publication. We have taken into consideration all the suggested reviews. In this letter we wish to address all the changes that were made to the subjected manuscript, in hope that they are optimal and the article will be accepted for publication. There will be two versions of the manuscript, one tracked changed and one clean for all changes. The answers will be provided under each suggestion in the given order.
I have a major concern and I believe the manuscript should be re-written according to this concern:
- The classification of MS has been changed and the authors should use the newest classification which divided MS to relapsing and progressive forms.
Response:
We thank the reviewer for the careful analysis of the manuscript and for valuable suggestions. Throughout the manuscript, the term secondary-progressive multiple sclerosis (SPMS) has been changed into progressive multiple sclerosis (PMS), as per the new definition of MS Phenotype Group in 2014. Inside the tables, where the studies of other authors are being presented, the term secondary-progressive has been kept as per their original source. Furthermore, new data (and references) have been added regarding the classification in the following paragraphs:
- Page 2 [72-75]:
In 2014, according to the new definition by MS Phenotype Group, MS was classified as relapsing-remitting and progressive disease. The early identification of the MS phenotype in which the predominant pathophysiological process is either inflammation (in RRMS) or neurodegeneration (in PMS) has a substantial impact on the therapeutical decision and the prognosis [18-19].
- According to studies neurodegeneration is ongoing process which begins at the onset of disease. So, term of transitional stage of this disease is old and it should be revised.
Response:
Transition stage refers to the moment when relapsing-remitting multiple sclerosis converts to progressive multiple sclerosis. Neurodegeneration is present in all stages of the disease, including the preclinical stage, but in progressive multiple sclerosis, neurodegeneration is the main pathophysiological mechanism. This data has been presented in the first two paragraphs of the manuscript:
- Page 1-2 [41-70]
“[…] However, recent data have confirmed that neurodegeneration is present in all clinical phenotypes, including early disease. Therefore, it was hypothesised that neurodegeneration and axonal loss are driven by independent pathological processes occurring before or concomitant to demyelination [1-6].
In 85–90% of the cases, MS presents a relapsing-remitting course and is defined as relapsing-remitting MS (RRMS). RRMS is characterised by episodic occurrence of symptoms with complete or partial recovery followed by periods of clinical stability and remission. According to the natural history data of MS, 25% of RRMS patients eventually convert to progressive MS (PMS) after one decade of evolution. This progression is characterised by a gradual decline in neurological function with or without superimposed relapses. Due to recent developments in potent disease-modifying therapies (DMTs) and high-accuracy diagnostic protocols, the percentage of patients who transition to a progressive phase has decreased. Nonetheless, two-thirds of RRMS patients will progress to PMS after 20 years of disease evolution. The conceptual de-lineation between RRMS and PMS is not clearly defined; the evolution is gradual, with an intermediate phase of ‘transitional MS’. Compensatory mechanisms may initially protect against neuronal injury; however, in time, their continuous decline progresses to exhaustion and irreversible neurological deficits, marking the transition to PMS. However, the presence of these compensatory mechanisms represents one of the main obstacles in early PMS diagnosis. MS patients have a substantial heterogeneity concerning the brain reserve, modifiable cognitive reserve and neuroplasticity [7-18].
In recent years, eminent researchers in the field of multiple sclerosis deployed a wide research arsenal aiming to identify the moment of transition from a relapsing-remitting phenotype to a progressive disease:
- Lublin et al. (2014) - Lublin, F.D.; Reingold, S.C.; Cohen, J.A.; Cutter, G.R.; Sørensen, P.S.; Thompson, A.J.; Wolinsky, J.S.; Balcer, L.J.; Banwell, B.; Barkhof, F.; Bebo Jr, B.; Calabresi, P.A.; Clanet, M.; Comi, G.; Fox, R.J.; Freedman, M.S.; Goodman, A.; D.; Inglese, M.; Kappos, L.; Kieseier, B. C.; Lincoln, J.A.; Lubetzki, C.; Miller, A.E.; Montalban, B. C.; O’Connor, P.W.; Petkau, J.; Pozzilli, C.; Rudick, R.A.; Sormani, M.P.; Stüve, O.; Waubant, E.; Polman, C.H. Defining the clinical course of multiple sclerosis: the 2013 revisions. Neurology 2014, 83(3), 278–286.
- Delphi Panel (Consensus for the Early Identification of Secondary Progressive Multiple Sclerosis) - Sá, M.J.; Basílio, C.; Capela, C.; Cerqueira, J.J.; Mendes, I.; Morganho, A.; Correia de Sá, J.; Salgado, V.; Martins Silva, A.; Vale, J.; Sousa, L. Consensus for the Early Identification of Secondary Progressive Multiple Sclerosis in Portugal: a Delphi Panel. Acta Med Port 2023, Online ahead of print.
- Holm et al. (2023) - Holm RP, Pontieri L, Wandall-Holm MF, Framke E, Sellebjerg F, Magyari M. The uncertainty period preceding the clinical defined SPMS diagnosis and the applicability of objective classifiers - A Danish single center study. Mult Scler Relat Disord. 2023 Feb 3;71:104546.
- Ziemssen et al. (2022) - Ziemssen T, Bhan V, Chataway J, Chitnis T, Campbell Cree BA, Havrdova EK, Kappos L, Labauge P, Miller A, Nakahara J, Oreja-Guevara C, Palace J, Singer B, Trojano M, Patil A, Rauser B, Hach T. Secondary Progressive Multiple Sclerosis: A Review of Clinical Characteristics, Definition, Prognostic Tools, and Disease-Modifying Therapies. Neurol Neuroimmunol Neuroinflamm. 2022 Nov 22;10(1):e200064.
- The biomarker related to transitional phase are not useful.
Response:
During the past 5 years, there is increasing evidence stated in numerous studies that there is an urgent need for transition to progressive multiple sclerosis biomarkers. According to Lublin et al. (2014), imaging techniques such as OCT represent a viable option for progression biomarkers, due to the ease of administration and lack of invasiveness.
- Page 2-3 [104-163]
According to Lublin et al. (2014), the diagnosis of PMS can be established only retrospectively and the authors underline the lack of clinical, imaging, immunological and pathological biomarkers for the transition point from RRMS to PMS and the pressing need to identify biological biomarkers that can enhance the MS phenotype definition. The authors suggested using optical coherence tomography (OCT) as an indirect biomarker for whole brain tissue loss and the importance of additional studies aiming to extend the indications for OCT as a potential biomarker of MS phenotype. They suggest prospective follow-up for MS patients and their clinical and paraclinical parameters that should be prioritised to understand the disease’s subtypes better and to accurately predict the transition moment between RRMS and PMS [18].
Ziemssen et al. (2023), in a recent article published in Neuroimmunology and Neuroinflammation, entitled Secondary Progressive Multiple Sclerosis pointed out that early identification of the transition phase between the disease phenotypes is essential since novel molecules for progressive multiple sclerosis are emerging.
Sá et al. (2023), as part of the Delphi Panel for early identification of secondary progressive multiple sclerosis in Portugal, recognized that identification of progression is a clinical challenge that would be completed by the discovery of biomarkers and diagnostic tools that point to the transition phase.
- It is better to describe which biomarkers could predict the speed and amount of neurodegeneration in this patient.
Response:
Thank you for the insight. The transition phase is defined by the over-exhaustion of compensatory mechanisms and the over-expression of the neurodegenerative processes. This moment is variable from one patient to the other, due to the heterogeneity of the brain reserve, modifiable cognitive reserve and neuroplasticity. At the moment numerous studies, as presented, describe capable biomarkers for distinguishing between relapsing-remitting to secondary progressive disorders. Most of these biomarkers reflect neurodegenerative processes (cerebral atrophy, thinning of the retinal nerve fibre layer, etc.) but the possibility to quantify these variables in a time-sensitive manner is a challenge for future research. It is true that neurodegeneration in multiple sclerosis and any other neurodegenerative diseases is a sensitive matter. The aim of this review was to discuss the progress made in identifying those molecular and imaging biomarkers that have prognostic value in disability evolution and conversion from relapsing-remitting multiple sclerosis to secondary progressive multiple sclerosis. Multiple sclerosis is a highly heterogeneous disorder, and future studies are required in order to quantify the time impact that the biomarkers’ value holds.
We thank you for the valuable input and hope that the presented aspects have clarified the inquiries brought upon this manuscript. We have conducted a new English revision throughout the manuscript, and minor corrections were amended as can be noted in the tracked changes version.
Best regards,
Laura Barcutean and the rest of the authors

Reviewer 2 Report
The paper aimed to review molecular and imaging biomarkers in MS. The manuscript deals with a topic of great interest; it is thorough and well-written. The tables provide a concise synthesis of the studies available on the topic.
I have some minor comments. The section regarding the biomarker reported only a limited number of biomarkers, and I would suggest expanding the section with other biomarkers (i.e. CHI3L1 or CXCL13 etc.). Moreover, in the section on neurofilaments, I would suggest mentioning the NF heavy chain and its role.
Author Response
Esteemed reviewer,
On behalf of my co-authors and me, we would like to thank you for considering our article for publication. We have taken into consideration all the suggested reviews. In this letter, we wish to address all the changes that were made to the subject manuscript, in hope that they are optimal and the article will be accepted for publication. There will be two versions of the manuscript, one tracked changed and one clean for all changes. The answers will be provided under each suggestion in the given order.
The paper aimed to review molecular and imaging biomarkers in MS. The manuscript deals with a topic of great interest; it is thorough and well-written. The tables provide a concise synthesis of the studies available on the topic.
Response:
Thank you for your support.
I have some minor comments. The section regarding the biomarker reported only a limited number of biomarkers, and I would suggest expanding the section with other biomarkers (i.e. CHI3L1 or CXCL13 etc.). Moreover, in the section on neurofilaments, I would suggest mentioning the NF heavy chain and its role.
Response:
Thank you for your valuable input. As suggested, we have updated the manuscript (and the figure accordingly) with recent data regarding the abovementioned biomarkers (CHI3L1, CXCL13 and neurofilaments heavy chains). Please find the following amendments performed on the manuscript:
- Neurofilaments heavy chain – page 5 (303-320)
Numerous studies demonstrated the utility of NfL as a prognostic biomarker in MS but in contrast, neurofilaments heavy (NfH) and phosphorylated NfH have been understudied. NfH and pNfH are cytoskeleton proteins expressed by the neurons in reaction to injury. They are of interest in neurodegenerative disorders such as amyotrophic lateral sclerosis, Alzheimer’s disease and frontotemporal dementia. Still, they have been studied to a lesser extent in MS. A recent study evaluated the role of serum/CSF NfH in MS patients according to OCT parameters. The authors demonstrated that SPMS patients had significantly higher NfH CSF levels and lower thickness of the peripapillary retinal nerve fibre layer (pRNFL) compared to RRMS patients and controls (p<0.0001). The pRNFL thickness positively correlated with serum and CSF NfH for both groups (p<0.01). The EDSS scores positively correlated with the CSF NfH levels both in RRMS and SPMS groups (p<0.001, p=0.002) [44].
The importance of NfH levels in MS patients is controverted. While some authors demonstrated that the serum and CSF NfH levels are higher in PMS patients compared to RRMS patients, others disproved any association of this biomarker with a disability, progression risk and imaging parameters [45-55].
Due to the lack of consistency in research results, the utility of NfH as a biomarker of transition in MS has yet to be established and further validation is needed.
- Chitinase-3-like-1 and Chemokine C-X-C motif ligand 13 - page 8-9 (539-603)
The exact role of chitinase-3-like-1 (CHI3L1) in MS pathophysiology has yet to be completely understood. Inside the central compartment, activated microglia and reactive astrocytes are the main producers of CHI3L1. Chemokine C-X-C motif ligand 13 (CXCL13) is a B lymphocyte chemoattractant involved in forming ectopic lymphoid follicles adjacent to the meninges.
Recent clinical studies demonstrated a significant association between the serum and CSF levels of CHI3L1 with disability accumulation, cognitive decline and disease progression in MS patients [90-94].
In a recent systematic review and meta-analysis, Floro et al (2022) evaluated the potential to use serum and CSF CHI3L1 levels as a biomarker for distinguishing dif-ferent MS phenotypes. Among the 20 studies researched over a period of 10 years (2010-2020), five compared the serum/CSF CHI3L1 levels in RRMS, SPMS and PPMS. They noted that PPMS patients are significantly higher compared to RRMS patients, but no differences were noted between RRMS and SPMS patients [95].
Lamancova P et al. (2022) evaluated serum levels of sNfL, CXCL13 and CHI3L1, among other molecular biomarkers and noted that they were significantly higher in PPMS and SPMS patients compared to RRMS. Furthermore, the CXCL13 levels corre-lated with the EDSS score. In the SPMS group, the authors noted a positive correlation between CHI3L1 and sNfL [96].
Similar results are reported by Gil-Perontin et al. (2019). The authors analysed CSF levels for NfL and CHI3L1 in MS patients. The CSF CHI3L1 levels were lower in RRMS patients compared to SPMS and PPMS patients but reached statistical signifi-cance only between RRMS and PPMS patients. In the same study, the CSF CHI3L1 positively correlated with the EDSS score and proved to be an independent factor of prediction for EDSS worsening by 1 point (HR=2.99, 95% CI 1.27-7.07). The patients were followed for 10 years, and 14 patients with RRMS converted to SPMS. The statis-tical analysis identified the CHI3L1 levels as a predictive biomarker for conversion. The results of these studies support the hypothesis that NfL, CHI3L1, and CXCL13 can be used to identify different MS phenotypes and have the potential to become predictive biomarkers of transition from RRMS to SPMS [97].
Two types of parallel inflammation are specific to MS. The first type is typical for the RR forms and is characterised by B and T lymphocyte passage through the vul-nerable BBB, with the formation of white matter demyelinating lesions. The second type of inflammation, although present from the early stages of the disease, defines compartmentalised inflammation in the presence of an impenetrable BBB. This results in the formation of tertiary ectopic lymphatic follicles adjacent to the meninge and the perivascular Virchow Robin spaces and is associated with cerebral and cerebellar sub-pial demyelinating lesions [98].
The results of the abovementioned studies suggest that an increase in CHI3L1 production is associated with the second type of inflammation. This hypothesis was confirmed by the study of Cubas-Núñez L et al. (2021), who analysed the CSF CHI3L1 levels in different MS patients and evaluated the CHI3L1 expression and the degree of inflammation and neurodegeneration found in the autopsy tissue in 22 MS patients. The CSF CHI3L1 level was significantly lower in PMS patients compared to RRMS and, furthermore, correlated with the EDSS score. At the level of chronic active lesions, CHI3L1 is mainly expressed by astrocytes and is associated with neurodegeneration [99].
The results of these studies suggest that NfL, CHI3L1 and CXCL13 can be used to identify different MS phenotypes but can also be used as predictive biomarkers for the moment of transition to a PMS. Still, this hypothesis has to be confirmed in a larger cohort.
We thank you for the valuable input and hope that the presented aspects have clarified the inquiries brought upon this manuscript. We have conducted a new English revision throughout the manuscript, and minor corrections were amended as can be noted in the tracked changes version.
Best regards,
Laura Barcutean and the rest of the authors

Reviewer 3 Report
Two-thirds of RRMS would develop to SPMS. Since the line between RRMS and SPMS is ambiguous, there are a few difficulties in the diagnosis of SPMS. Consequently, the diagnosis of SPMS usually have 3-year delay, which could delay the therapeutic adjustment. The authors review potential biomarkers that could facilitate early identification of RRMS patients who are at risk of progressing to SPMS. The biomarkers were classfied into molecular biomarkers (NfL, circulating microRNA, GFAP, kynurenines), OCT biomarkers, MRI biomarkers. The comprehensive review assisted improve our knowledge of SPMS biomarker, and guide our clinical practice.
Author Response
Esteemed reviewer,
On behalf of my co-authors and me, we would like to thank you for considering our article for publication.
Two-thirds of RRMS would develop to SPMS. Since the line between RRMS and SPMS is ambiguous, there are a few difficulties in the diagnosis of SPMS. Consequently, the diagnosis of SPMS usually have 3-year delay, which could delay the therapeutic adjustment. The authors review potential biomarkers that could facilitate early identification of RRMS patients who are at risk of progressing to SPMS. The biomarkers were classfied into molecular biomarkers (NfL, circulating microRNA, GFAP, kynurenines), OCT biomarkers, MRI biomarkers. The comprehensive review assisted improve our knowledge of SPMS biomarker, and guide our clinical practice.
We thank you for your valuable input and support and hope that the presented aspects, with the modifications that are clarified in the revised manuscript add value to our research. We have conducted a new English revision throughout the manuscript, and minor corrections were amended as can be noted in the tracked changes version.
Best regards,
Laura Barcutean and the rest of the authors

Round 2
Reviewer 1 Report
Thanks a lot